

# Climate and ocean circulation changes toward a modern snowball Earth

Takashi Obase[1], Takanori Kodama[2], Takao Kawasaki[3], Sam Sherriff-Tadano[4], Daisuke Takasuka[5], Ayako Abe-Ouchi[3], and Masakazu Fujii[6]

[1]Japan Agency for Marine-Earth Science and Technology, Showa-machi, Kanazawa, Yokohama, Kanagawa, 236-0001, Japan
[2]Earth-Life Science Institute (ELSI), Institute of Science Tokyo, I7E-315, 2-12-1, Ookayama, Meguro, Tokyo, 152-8550, Japan.
[3]Atmosphere and Ocean Research Institute, The University of Tokyo, Kashiwa, Japan
[4]The University of Ryukyus, Nishihara, Japan
[5]Department of Geophysics, Tohoku University, Sendai, Japan
[6]National Institute for Polar Research, Tachikawa, Japan

**Correspondence:** Takashi Obase (tobase@jamstec.go.jp)

**Abstract.** In the past, Earth experienced snowball events, where its surface became completely covered with ice. Previous studies used general circulation models to investigate the onset and climate of such snowball events. Using the MIROC4m coupled atmosphere–ocean climate model, this study examined the changes in the oceanic circulation during the onset of a modern snowball Earth and elucidated their evolution to steady states under the snowball climate. Abruptly changing the solar constant to $94\%$ of its present-day value caused the modern Earth climate to turn into a snowball state after 1300 years and initiated rapid increase in sea ice thickness. During onset of the snowball event, extensive sea ice formation and melting of sea ice in the mid-latitudes caused substantial freshening of surface waters and salinity stratification. By contrast, such salinity stratification was absent if the duration necessary for snowball onset was short because of stronger solar constant forcing. After snowball onset, the global sea ice cover reduced air–sea fluxes and caused drastic weakening in the deep ocean circulation. However, as the ocean temperature and salinity fields approached near constant states, the meridional overturning circulation resumed in the steady-state snowball climate. Although the evolution of the oceanic circulation would depend on model setting, particularly regarding the treatment of air–sea fluxes and the continental distribution, our results highlight the importance of the oceanic circulation and associated biogeochemical changes in the climate system feedback and sequence of snowball events.

## 1 Introduction

Studies have suggested that Earth has experienced major glaciation episodes, and that its surface was covered with ice during the Cryogenian Period (720–635 Ma) and the Paleoproterozoic Era ($\sim 2.4$ Ga) (Hoffman et al., 2017). The snowball Earth hypothesis is supported by multiple sources of evidence such as glacial deposits in low-latitude regions (Harland, 1964), iron formation, and cap carbonates (Kirschvink, 1992; Kirschvink et al., 1999; Hoffman et al., 2017). The climate forcing and the distribution of continents during past snowball Earth events differed from those of the present-day Earth. Solar luminosity is estimated to have been approximately $94\%$ that of its present-day value during the latter stage of the Cryogenian Period,





and several boundary conditions for climate, including the continental distribution and the concentrations of atmospheric greenhouse gases, were changing over time and differed from the current situation. Thus, it is important to address why snowball Earth events occurred during the Paleoproterozoic Era and the Cryogenian Period but not since. Climate system modelingcan provide the opportunity both to understand the dynamics and interactions of climate system components, and to

examine the conditions necessary for the initiation, maintenance, and termination of a snowball Earth event.

One-dimensional, latitudinal energy balance models (EBMs) of the Earth have shown that the snowball state is one equilibrium solution of the planet as a function of incoming solar flux because of the presence of ice–albedo feedback (Budyko, 1969; Sellers, 1969). Recent studies have used atmospheric general circulation models (AGCMs) to produce planetary solutions in response to insolation changes, which have different distributions of surface water with conditions that range from a snowball

climate to a runaway greenhouse effect (Abe et al., 2011; Rose, 2015; Yang et al., 2017; Kodama et al., 2021). While climate models typically do not consider changes in continental ice sheets, studies involving asynchronous simulation of an ice sheet model with an EBM indicate that the ice sheet has positive feedback in the snowball solution (Hyde et al., 2000; Peltier et al., 2007; Liu and Peltier, 2010).

Three-dimensional ocean dynamics also impact the snowball conditions. Studies using coupled atmosphere–ocean general

circulation models (AOGCMs) have shown that ocean and sea ice dynamics affect the thresholds of snowball conditions via meridional transport of sea ice and albedo feedback (Voigt and Marotzke, 2010; Yang et al., 2012a, b, c; Liu et al., 2018). Such studies indicated that the threshold of the incoming solar flux necessary for the onset of a snowball event in the modern Earth configuration is 89.5–92 %. Subsequent studies used the same AOGCM to investigate the threshold for snowball onset under the past configuration of the Marinoan (720–660 Ma) and Sturtian (650–630 Ma) periods based on reconstructions of

the continental distribution (Poulsen et al., 2002; Voigt et al., 2011; Voigt and Abbot, 2012; Voigt, 2013; Liu et al., 2013; Eberhard et al., 2023). It was demonstrated that the necessary change in incoming solar flux for the onset of a snowball event during the Marinoan or Sturtian periods was smaller than that of the present-day configuration, indicating that initiation of a snowball event would have been easier (Voigt et al., 2011; Liu et al., 2013). On the basis of the output of a two-dimensional EBM coupled with an ice sheet model (Hyde et al., 2000; Peltier et al., 2004) or an AOGCM (Abbot et al., 2011; Yang et al.,

2012a), it has been suggested that a slushball is one planetary steady state that is characterized by only a small area free of sea ice in tropical regions.

The oceanic circulation during the snowball state and its changes during snowball onset and termination are important for understanding biogeochemical processes and marine sediment records. Ashkenazy et al. (2014) used a three-dimensional ocean general circulation model (OGCM) coupled with a sea–glacier model with surface atmospheric conditions from an

EBM (Pollard and Kasting, 2005). They found a meridional circulation characterized by upwelling in the tropics driven by geothermal heat flux and the gradient in sea ice thickness. Ramme and Marotzke (2022) used an AOGCM and simulated snowball onset and termination by changing the concentration of atmospheric $CO_2$, and they continued the snowball initiation experiments after 150 years from snowball onset to determine the initial conditions of a deglaciation experiment. They showed that the salinity and temperature fields of the ocean approach near-uniform states because of the remaining meridional ocean





circulation (MOC). They also revealed the importance of the atmospheric circulation in driving the oceanic circulation, which destabilized the salinity stratification after snowball termination.

As indicated above, the threshold of incoming solar flux or atmospheric greenhouse gas concentration for snowball onset has been quantified, as has the influence of the continental distribution. Ramme and Marotzke (2022) used an AOGCM to analyze the oceanic circulation after snowball deglaciation and discussed the implications for the geological record. However,

consideration of how the MOC might change over time from snowball onset to the establishment of a steady snowball state is lacking. Comprehension of the dynamics of the oceanic circulation under the conditions of a snowball state is essential for understanding the geochemical processes and elucidating their roles both in maintaining a snowball state for millions of years and in snowball termination.

In this study, we conducted steady-state experiments by reducing the solar flux under the modern continental configuration

using the MIROC AOGCM. Our experimental design was comparable to that of previous studies on the modern snowball Earth (Voigt and Marotzke, 2010; Yang et al., 2012a). We analyzed the changes in the oceanic circulation and the temperature and salinity fields, and investigated their evolution toward steady states. We applied minimal changes to the model regarding the treatment of sea ice dynamics and sea surface momentum flux. However, it remains challenging for climate models to formulate the sea surface momentum flux in snowball Earth states where the sea ice becomes much thicker than that of the

modern climate. For example, Pollard et al. (2017) turned off the air–sea momentum flux when the sea ice thickness exceeded 6 m. Ramme and Marotzke (2022) turned off the air–sea momentum flux in their snowball termination experiment, but they used modern climate settings in their snowball initiation experiments.

We note that the conditions for snowball onset depend on several boundary conditions such as continental distribution, solar flux forcing, and atmospheric greenhouse gas forcing. Additionally, the simulated pattern of the ocean general circulation is

also affected by several boundary conditions such as the continental distribution, seabed topography, and geothermal heat flux. While these differences in the boundary conditions of our experimental design limited direct comparison with past snowball Earth events, our simulation of the modern snowball Earth elucidated the physics of the atmosphere–ocean dynamics (Voigt and Marotzke, 2010; Yang et al., 2012a), and supported investigation of the drivers of the oceanic circulation.

The remainder of this paper is structured as follows. In Sect. 2, we describe our climate model and explain the code changes

made to the ocean model to permit thick (several hundreds of meters) sea ice in MIROC. In Sect. 3, we present our results. Finally, Sect. 4 presents discussion regarding the limitations of our model and experiments, together with additional sensitivity experiments performed to investigate the influence of uncertain parameters related to the atmospheric boundary layer at the sea surface. Section 4 presents discussions on implications in relation to the climate system of snowball Earth.

## 2  Methods

### 2.1  Model

We used the MIROC4m AOGCM (Hasumi and Emori, 2004), which contributed to the historical and future projections of climate used in the fourth assessment report of the Intergovernmental Panel on Climate Change. The MIROC4m AOGCM



consists of the CCSR-NIES AGCM (horizontal resolution is T42 ($2.8°$) with 20 vertical levels), as in Abe et al. (2011) and Kodama et al. (2021), coupled with the COCO ocean general circulation model component (horizontal resolution is $1.4° \times$
$1.0°$ with 43 vertical levels), as in Oka et al. (2011). MIROC4m contributed to the Paleoclimate Modeling Intercomparison Project from the second phase to the most recent fourth phase (Braconnot et al., 2006; Kageyama et al., 2021), and it has been used in simulation of the paleoclimates of glacial periods (Abe-Ouchi et al., 2013; Obase and Abe-Ouchi, 2019; Sherriff-tadano et al., 2020; Kuniyoshi et al., 2022) and past warm climates up to the Cretaceous with vegetation feedback (O'ishi et al., 2021; Higuchi et al., 2021). The equilibrium climate sensitivity of MIROC4m, defined by the global mean surface air temperature
(SAT) with a doubling of the preindustrial level of atmospheric $CO_2$, is 3.9 K (Chan and Abe-Ouchi, 2020).

A previous study identified that the albedos of snow and ice are critical model parameters regarding the threshold of snowball onset (Yang et al., 2012a). We adopted the standard albedo parameters used for present-day climate and paleoclimate simulations to ensure consistency with previous studies that used MIROC4m. The shortwave albedo of sea ice and ice sheets was set at a constant value of 0.5. The shortwave albedo of snow was defined as a function of temperature to parameterize partial snow
cover at relatively high temperatures, i.e., the albedo was set at a value of 0.85 for temperatures of $-5$ °C or colder, and it was reduced linearly to a value of 0.65 for temperatures up to 0 °C. The COCO ocean model adopts a dynamic and thermodynamic sea ice model. Therefore, the maximum sea ice concentration was set as a function of sea ice thickness (linear function from 0.95 to 1 for ice thickness of 0 to 12 m) to parameterize sea ice leads. The salinity of sea ice was set to 5 psu.

## 2.2 Modification in the ocean model

The COCO ocean model uses a hybrid vertical coordinate, where the top 8 layers (45 m) are represented by sigma layers to represent sea surface height changes and the bottom layers are represented by $z$ coordinates. The formation of sea ice reduces the local sea surface height because freshwater is extracted from the ocean and incorporated in the ice during sea ice formation. This means that the sea ice thickness in a snowball climate (greater than $\sim 45$ m) leads to crashing of the ocean model because the sea surface height becomes lower than the thickness of the sigma layers. To avoid this problem, we changed
the code to conserve the local sea surface associated with sea ice formation or melting. Meanwhile, the calculated ocean salinity change associated with sea ice formation or melting was considered. We incorporated this change to ensure minimal difference in the climate state of the present-day simulation. Under this setting, both global water volume (sum of seawater and sea ice) and total salinity content were no longer preserved. Nevertheless, the effect of brine rejection associated with sea ice formation and freshening associated with sea ice melting were represented as in the original model. As described in Sect. 3, we
obtained comparable present-day sea ice distributions and meridional overturning circulations in the preindustrial simulations as generated in the original MIROC4m simulations.

## 2.3 Experimental design

All the experiments were initialized with the steady-state of the preindustrial simulations. The orbit of the Earth and the atmospheric greenhouse gas concentrations ($CO_2$ concentration: 285 ppm) were set to pre-industrial values, and the solar
constant was set at $1366.12$ W/m$^2$. In experiments TSI091–TSI100, the solar constant was changed from 91–100 % of the





preindustrial value, as in previous modern snowball studies (Voigt and Marotzke, 2010; Yang et al., 2012a). It should be noted that the geothermal heat flux at the seafloor was set to zero, as in the preindustrial and paleoclimate simulations.

In MIROC4m, the land surface vegetation type based on present-day biomes is given as an external boundary condition and remains fixed. The distribution of the continental ice sheet is also fixed to that of the present-day. However, because retaining

the present-day land surface vegetation is unrealistic in a snowball climate, all land surface types were abruptly replaced with ice sheets when the sea ice covered the global ocean. We found that experiments TSI094 and TSI091 were below the threshold for snowball onset. Therefore, the replacement of land surface types was conducted at year 430 in the TSI091 experiment and at year 1330 in the TSI094 experiment (Fig. 1). The TSI094 experiment was continued for more than 2000 years after snowball onset and the total simulation ran for 3500 years (Table 1).


| Name | Solar constant | Simulation |
|--------|--------|--------|
| TSI100 | 100% (1366.12 $W/m^2$) | 1000 years from the present-day climate |
| TSI096 | 96% | 2000 years from the present-day climate |
| TSI094 | 94% | 3500 years from the present-day climate |
| TSI091 | 91% | 1000 years from the present-day climate |

Table 1: List fo experiments in this study.

## 3   Results

### 3.1   Onset of snowball climate

Fig 1 shows the time series of the global mean sea ice area, sea ice thickness, and annual SAT for all four experiments. We did not realize the snowball state in experiment TSI096 (solar constant set at 96 % of the present-day value) within 2000 years (yellow line), indicating that the solar constant threshold for snowball onset was below 96 %. By contrast, when the solar constant was set to 94 % of the present-day value (TSI094), the entire ocean became covered with sea ice after approximately 1330 years (green line). Therefore, the solar constant threshold for a modern snowball climate in MIROC4m was determined

to be between 94 and 96 %. The solar constant required for snowball initiation in our study was larger than that reported in previous studies with modern configurations (Voigt and Marotzke, 2010; Yang et al., 2012a, b, c; Liu et al., 2018), suggesting greater sensitivity to insolation forcing in MIROC. Additionally, the sea ice area just before entering the snowball phase was approximately 40 %, i.e., smaller than that reported in the previous studies listed above, suggesting stronger ice–albedo feedback in MIROC. At the end of the TSI094 experiment, after 2000 years from snowball onset, the global mean sea ice

thickness was close to 250 m with a trend of increase (Fig. 1b). Although the time required for snowball onset was different between the TSI091 and TSI094 experiments, the rates of sea ice thickening and fall in global mean SAT after snowball onset





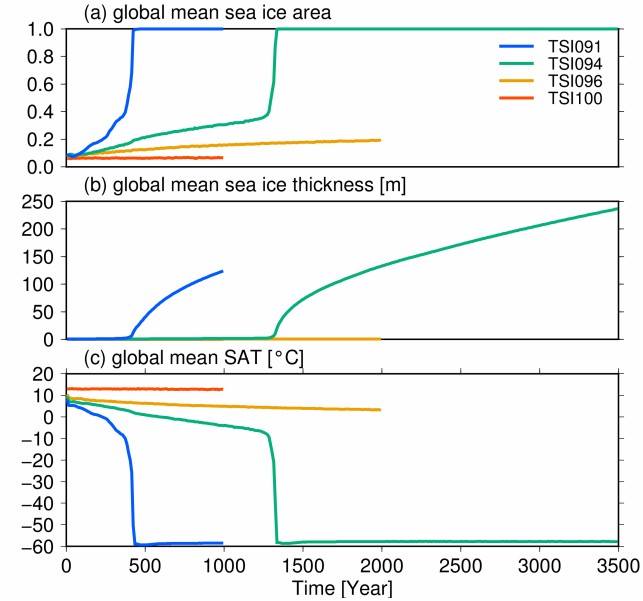

**Figure 1.** Time series of (a) global mean sea ice covered area, (b) global mean sea ice thickness, and (c) global mean surface air temperature (SAT).

were the same (Fig. 1c). Moreover, the simulated sea ice growth rate of both experiments was similar to that of Ramme and Marotzke (2022), realizing a sea ice thickness of 60 m 150 years after snowball onset.

Figure 2 shows time series of the volume transport of the wind-driven subtropical shallow MOC and the deep MOC (asso-
ciated with the Antarctic Bottom Water, AABW) cell. As in Voigt and Marotzke (2010), the subtropical shallow MOC and the AABW cell both strengthened before the onset of snowball transition in the TSI091 and TSI094 experiments. However, certain differences exist between the MOC time series of the TSI091 and TSI094 experiments. The first difference is the overshoot of the AABW cell just before snowball onset. The strength of the AABW cell reached a value of approximately 100 Sv in the TSI091 experiment, whereas no such overshoot occurred in TSI094 and the strength of the AABW cell remained at ap-
proximately 50 Sv (Fig. 2c blue). The second difference is the weakening after snowball onset. In the TSI091 experiment, the strength of the AABW cell fell to a minimum value of approximately 20 Sv some 300 years after snowball onset, whereupon it stabilized. In the TSI094 experiment, the strength of the AABW cell fell to less than 1 Sv approximately 300 years after snowball onset, following which it gradually recovered to reach a flow rate of 20 Sv (similar strength to that in TSI091) some 1000 years after snowball onset.

Figure 3 shows the evolution of the sea ice extent and global MOC in the TSI094 simulation across snowball onset, from partial coverage of sea ice to full snowball conditions. The transition, characterized by the migration of the sea ice edge from the mid-latitudes to tropical regions, was very rapid (Fig. 3a,b), as reported in previous studies (Donnadieu et al., 2004; Marotzke and Botzet, 2007; Voigt and Marotzke, 2010). The streamfunction of the AABW cell was greater than that of the modern climate (Fig. 3d,e) owing to the greater production of sea ice in the Southern Hemisphere than in the Northern Hemisphere.





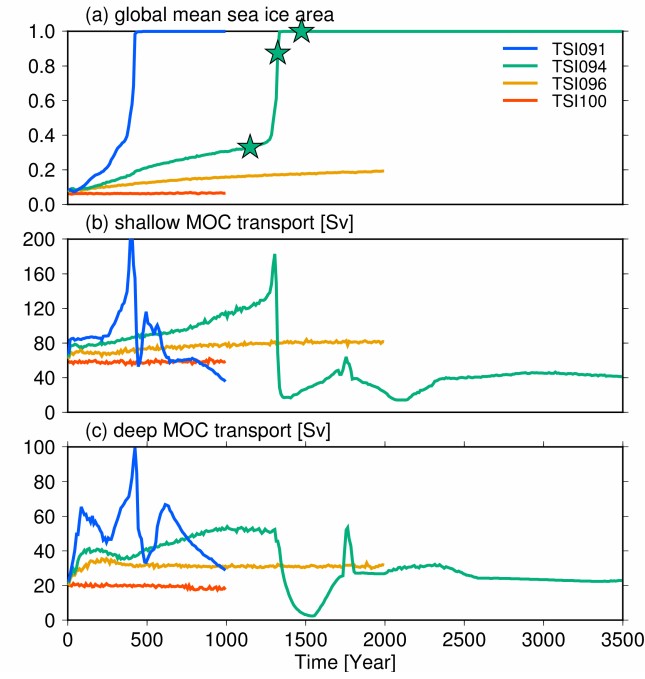

**Figure 2.** Time series of (a) the global mean sea ice area (as in Fig. 1), (b) the wind-driven subtropical shallow MOC (sea surface to depth of 1000 m), and (c) the deep MOC cell (depth of 3000 m to the seafloor) corresponding to AABW cell. Green stars in (a) represent snapshot states depicted in Fig. 3.

The AABW cell remained present during snowball onset. However, after snowball onset, the MOC declined rapidly and the strength of the deep ocean circulation associated with the AABW cell fell to less than 1 Sv (Fig. 2c and Fig. 3f).

### 3.2   Climate system of snowball Earth: Ocean

We compared the sea surface climate states and ocean circulation patterns from the three experiments corresponding to the states of the modern climate (TSI100), above the snowball threshold (TSI096), and snowball Earth (TSI094). We also compared
the 100-year climatology at the end of the respective experiments (Table 2.3), i.e., at year 1000 and 2000 in the TSI100 and TSI096 experiments, respectively. Although the sea ice was still increasing at the end of the TSI094 simulation, we analyzed the 100-year climatology at year 3500 from the present-day initial condition. The control simulation (TSI100) largely reproduced the present-day sea ice extent based on observations (Fig. 4a). In TSI096, the sea ice expanded in the high latitudes of both hemispheres but the global sea ice area remained at 20 % (Fig. 1a). Expansion of the sea ice area in TSI096 contributed to
increase in the surface albedo over the ocean (Fig. 4b middle). In the TSI094 experiment, the global ocean was totally covered with sea ice and snow cover, which resulted in an albedo of approximately 0.85 (Fig. 4b right). By contrast, the albedo over the continent was notably smaller than that of the ocean. The smaller albedo over the continent can be attributed to the absence



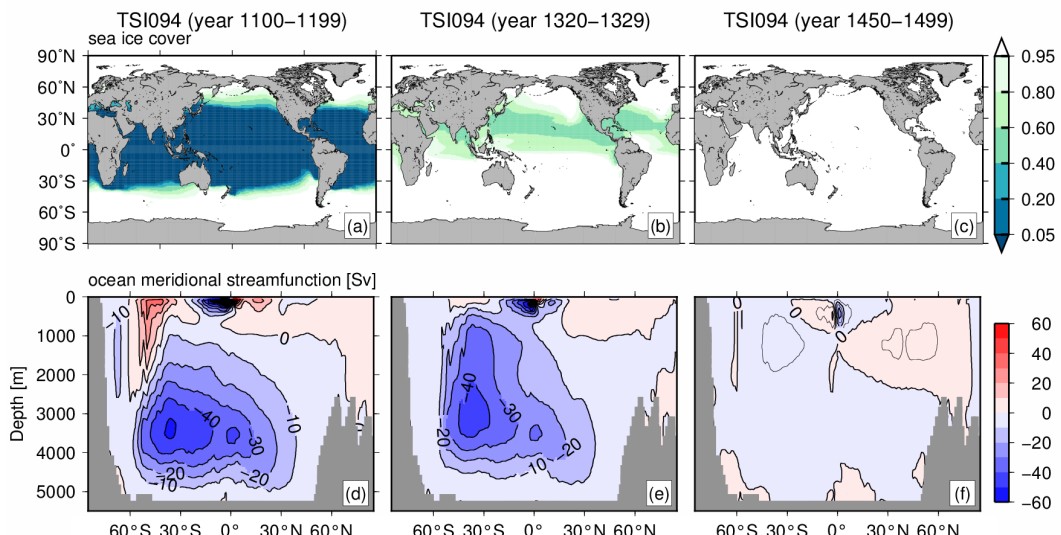

**Figure 3.** Sea ice extent and oceanic meridional streamfunction (positive indicates clockwise circulation) at three selected time-slice snapshots from the TSI094 experiment corresponding to the green stars shown in Figure 2(a).

of snow cover over the land and the surface albedo of the ice sheet (0.5). Evaporation exceeding precipitation was the reason for this surface albedo pattern over land in the TSI094 experiment, the causes of which are discussed later.

The present-day climate is characterized by cells of North Atlantic Deep Water (NADW) and AABW, as in the TSI100 simulation (Fig. 5a). The NADW cell was weakened substantially in the TSI096 case by expansion of sea ice in the North Atlantic (Fig. 4a middle) which prevented deep water formation. By contrast, extensive cooling in the Antarctic Ocean promoted active sea ice production, and brine rejection led to increased AABW volume transport and more saline bottom water (Fig. 5b). Under weak salinity stratification, two symmetric deep MOC cells of $\sim 20$ Sv formed in both hemispheres in TSI094 (Fig. 5c). The

increase in global salinity in TSI094 was due to the notable increase in sea ice volume that induced brine rejection, as reported in Ashkenazy et al. (2014).

The sea ice area was mostly in equilibrium in the TSI100 and TSI096 simulations because net sea ice production in the polar region was compensated by sea ice transport away from the polar regions and sea ice melting in subpolar regions (Fig. 6a). The TSI094 experiment exhibited slightly positive sea ice formation in the global ocean because the sea ice thickened as a rate

of 0.1 m per year (Fig. 1b). The sea ice formation and brine rejection contributed to increase in global mean salinity (Fig. 5f), as reported in previous studies (Ashkenazy et al., 2014; Ramme and Marotzke, 2022). Moreover, the meridional overturning circulation was present with a vertically uniform salinity profile in the snowball state (Fig. 5f). Figure 6b shows the atmospheric eastward wind component at 10-m height over the ocean, characterized by westward trade winds in the tropic and subtopics, and eastward winds in the northern and southern high latitudes associated with the westerlies. In the snowball climate, the wind

was weakened markedly but the pattern of the trade winds and westerlies remained. The westward wind over the Pacific Ocean transported sea ice, which led to the maximum sea ice thickness along the western side of the Pacific Ocean (Fig. 4a right).



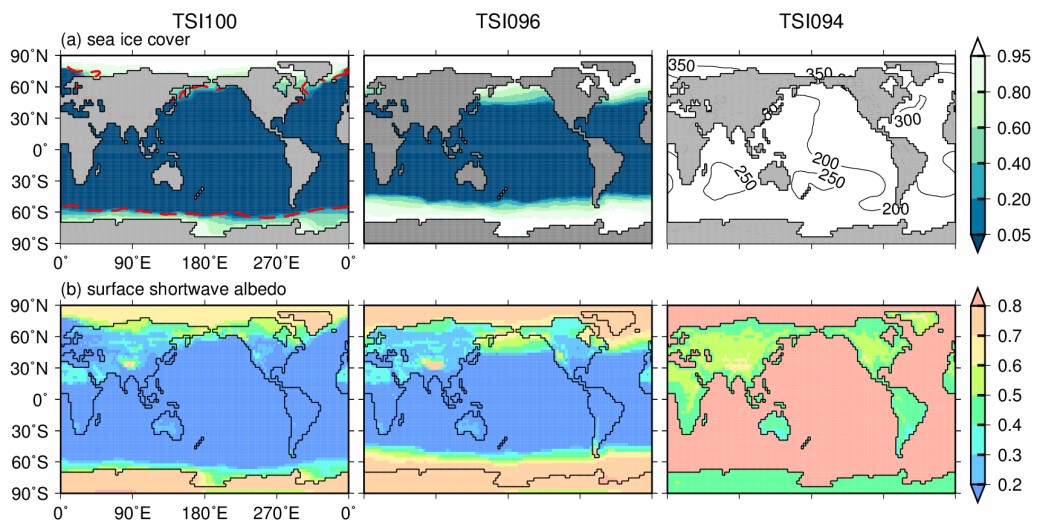

**Figure 4.** (a) Annual mean sea ice concentration and sea ice thickness at the end of the simulations. Red lines in TSI100 indicate the present-day winter sea ice edge, and the contour in TSI094 indicates the simulated annual mean sea ice thickness (unit: m). (b) Annual mean surface shortwave albedo.

The velocity of the sea ice was approximately 10 cm/s in TSI094, and the momentum of the sea ice was transmitted to the internal ocean. The deep ocean circulation and salinity distribution achieved in the TSI094 experiment can be interpreted as follows. While the sea surface salinity flux associated with sea ice growth (approximately 0.1 m per year) was broadly uniform in the global ocean (Fig. 6a right), the spatial distribution of the sea surface momentum flux associated with atmospheric winds induced upwelling in the tropics via convergence of surface currents (Fig. 6b). Even though the momentum flux was weaker than that of the modern climate, the broadly uniform ocean temperature and salinity might have contributed to the existence of the MOC. We note that the nearly uniform ocean salinity was also the result of the broadly uniform sea surface flux associated with sea ice production (Fig. 6a right) and the wind-driven ocean circulation that mixed the seawater.

## 3.3 Evolution of ocean circulations

Figure 7 shows the evolution of the zonal mean sea ice production in the TSI094 experiment. In the early phase of the experiment, sea ice formation occurred along the Antarctic coast and in the Arctic Ocean, as in the modern climate. The transport of sea ice to lower latitudes contributed to net sea ice melting in the Southern Ocean and the North Atlantic. The sea ice edge and the area of sea ice formation migrated to lower latitudes over time as the area of sea ice increased. Notably, the Southern Ocean in the region of the Antarctic Circumpolar Current at approximately 60°S, which is currently an area of substantial sea ice melting, became an area of active sea ice formation after 1000 years in the experiment. The melting of sea ice in the mid-latitudes also increased, contributing to reduction in the surface salinity. Generally, greater production of sea ice in the Southern Hemisphere than in the Northern Hemisphere contributed to the strong AABW cell prior to snowball onset (Fig. 3d,e)



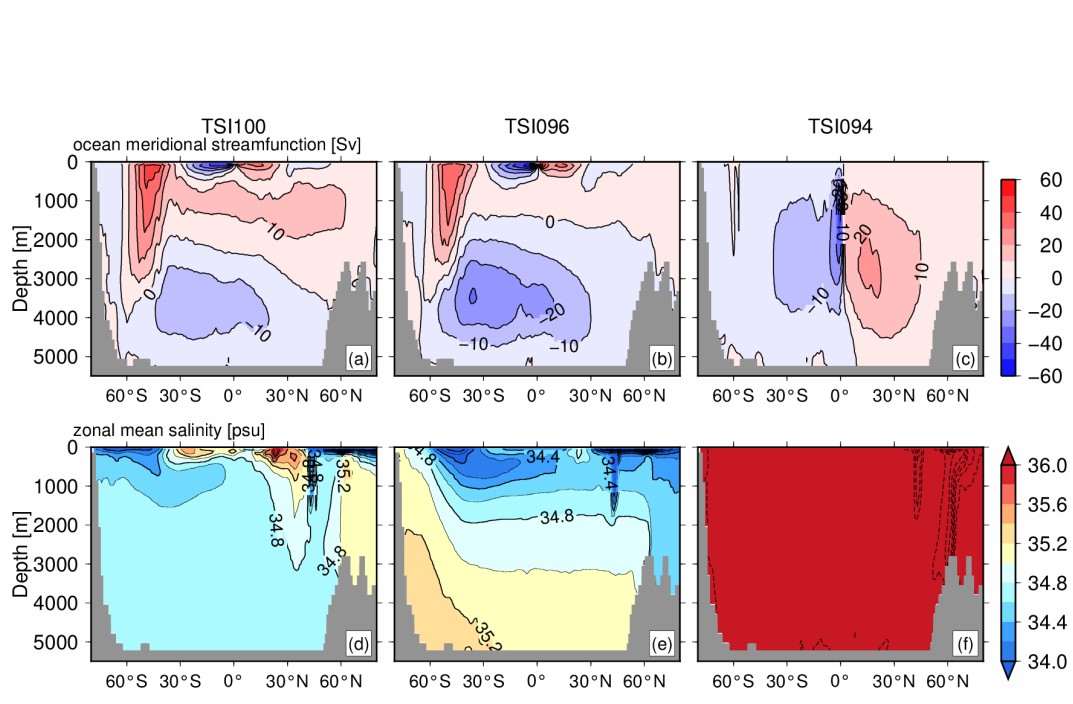

**Figure 5.** (a) Oceanic meridional overturning circulation (positive indicates clockwise circulation) and (b) zonal mean salinity in the three experiments.

It is also notable that the rapid expansion of sea ice before snowball onset was faster in the Southern Hemisphere than in the Northern Hemisphere.

Figure 8 compares Hovmöller diagrams of the global mean ocean temperature and the salinity profile of the TSI094 and TSI091 experiments. In the TSI094 experiment, the salinity of the surface layer decreased owing to increased melting of sea ice in mid- and low-latitude areas, while the salinity of the deep layer increased owing to sea ice production and brine rejection in the polar regions (Fig. 7). The global mean sea surface salinity dropped to 31.4 psu in TSI094 before snowball onset (Fig. 8a left). Conversely, the enhanced salinity stratification found in the TSI094 experiment was absent in the TSI091 experiment, where the global mean sea surface salinity at snowball onset was 33.8 psu. The temperature and salinity profiles approached uniform values after the snowball onset in both experiments, but TSI094 took longer to resolve the salinity stratification at snowball onset. Hence, the strong salinity stratification resulting from active sea ice melting just prior to snowball onset was found only in TSI094. The strong salinity stratification at snowball onset probably caused the strength of the AABW cell to weaken to approximately 1 Sv after snowball onset in TSI094 (Fig. 2c), while the strength of the AABW cell was maintained at a strong value in TSI091 where such strong salinity stratification was absent. The poleward oceanic heat transport became stronger toward snowball onset (Fig. 9). The increase in the meridional heat transport that occurred only in the Southern Hemi-



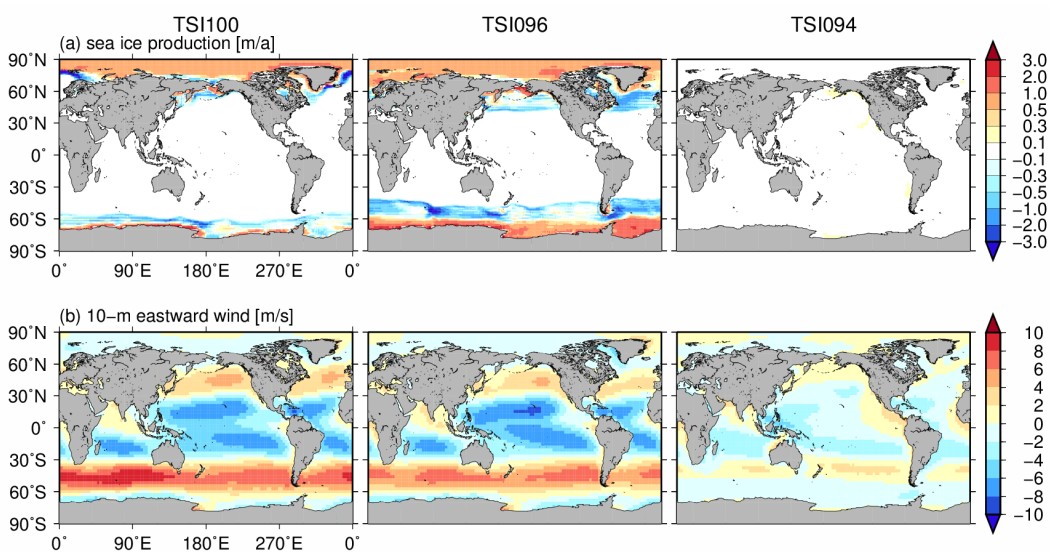

**Figure 6.** (a) Annual mean rate of sea ice production (freshwater equivalent m/s) and (b) eastward winds at 10-m height in the three experiments.

sphere, similar to the results of Voigt and Marotzke (2010), was due to strengthening of the AABW cell and the disappearance of the NADW. After snowball onset, the oceanic heat transport diminished rapidly as the ocean temperature approached uniformity at the freezing point, eliminating the temperature difference between low- and high-latitude areas (Fig. 8b).

### 3.4 Atmosphere of the snowball Earth climate as a driver of ocean circulations

We also analyzed the atmospheric temperature and circulations at the same period (Figs. 4–6) to elucidate the importance of air–sea fluxes as drivers of ocean circulations. The annual mean zonal mean air temperature of the snowball climate was 220 K in the tropics and 200 K in polar regions (Fig. 10a). The maximum monthly air temperature occurred in the summer over the Antarctic continent (Fig. 10b). The relatively warm Antarctic summer can be attributed to the smaller surface albedo over the Antarctic continent (Fig. 4b). The smaller temperature gradient between the tropics and the polar regions caused smaller meridional atmospheric and oceanic heat transport. The oceanic meridional heat transport was weakened in TSI096 (Fig. 11b) because of the absence of NADW formation (Fig. 5b), whereas in the snowball climate (TSI094 and TSI091), there was minimal oceanic heat transport (Fig. 11). The atmosphere still transported heat but the amount was reduced markedly relative to that of the modern climate. Compared with the findings of a multimodel study by Abbot et al. (2011), our snowball Earth results exhibited generally colder surface air temperatures, which can be explained by the different concentration of atmospheric $CO_2$ used, and the albedo of snow (0.85) in the snowball states (Fig. 4b right) compared with the constant value of 0.7.





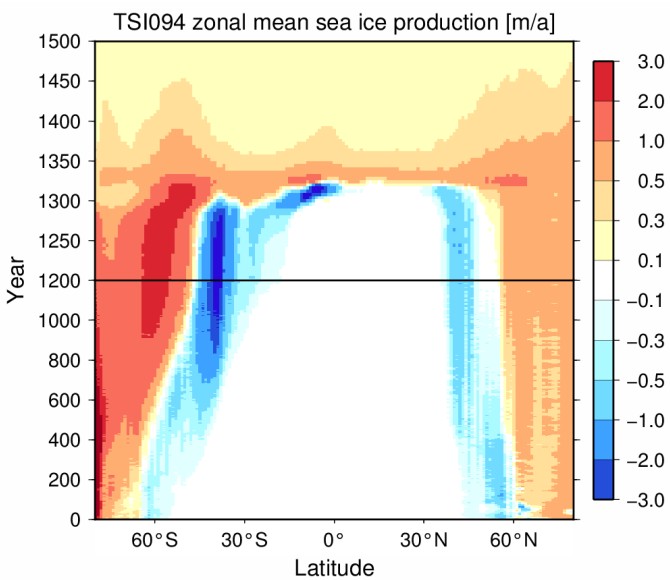

**Figure 7.** Time series of the zonal mean sea ice production (freshwater equivalent m/a) in the TSI094 experiment. Note that the vertical scale is changed at year 1200 to highlight the rapid transition to the snowball Earth climate.

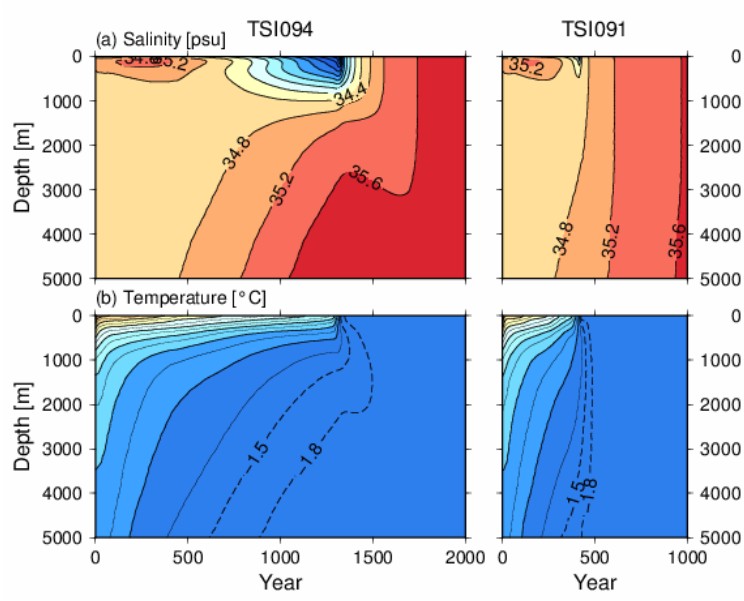

**Figure 8.** Time series of the vertical profile of the global mean ocean (a) salinity and (b) temperature in the TSI094 and TSI091 experiments.

Interestingly, the atmospheric circulations and clouds in our experiments differed from those of previous studies. First, our
245  simulations exhibited weaker Hadley circulations in the snowball Earth climate than found in the modern climate (Fig. 12).




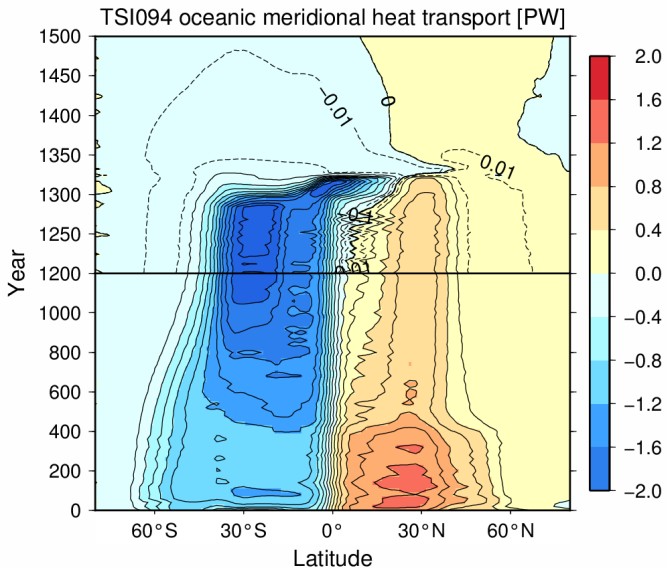

**Figure 9.** Time series of the oceanic meridional northward heat transport in the TSI094 experiment. Note that the vertical scale is changed at year 1200 to highlight the rapid transition to the snowball Earth climate.

The weaker Hadley circulation in the snowball climate differed from that in multimodel studies because they showed stronger seasonal Hadley circulations and reversed Hadley circulations in the annual mean (Abbot et al., 2013). Second, most models produce notable cloud fractions in the snowball Earth climates (Abbot et al., 2012); however, our simulation produced a very sparse cloud fraction (Fig. 13a bottom) and very weak cloud radiative forcing (Figure 13b bottom). Third, while multimodel studies produced net evaporation at the equator (Abbot et al., 2013), our model exhibited net precipitation over the ocean and sublimation from the land ice sheet (Fig. 14a left). Although our snowball climate states are generally consistent with those of previous studies regarding the zonal mean atmospheric air temperature and reduced temperature gradient, the atmospheric circulation, cloud fraction, and water cycles are very different. We have found that uncertain model parameterization of the turbulence coefficient at the sea surface was one of the reasons for the notable differences between the results of our experiments and the findings of previous AGCM studies. In Sect. 4, we conduct the additional sensitivity experiments to understand the cause of this difference.

## 4   Discussion

### 4.1   Threshold of snowball onset

On the basis of our MIROC experiments, we found that the threshold for snowball onset for the modern Earth is approximately 94 % of the present-day value of insolation. This threshold for snowball onset is higher than that derived in other AOGCM studies, e.g., 89.5–92 % (Voigt and Marotzke, 2010; Yang et al., 2012a, b, c)). One possible explanation for the greater sus-





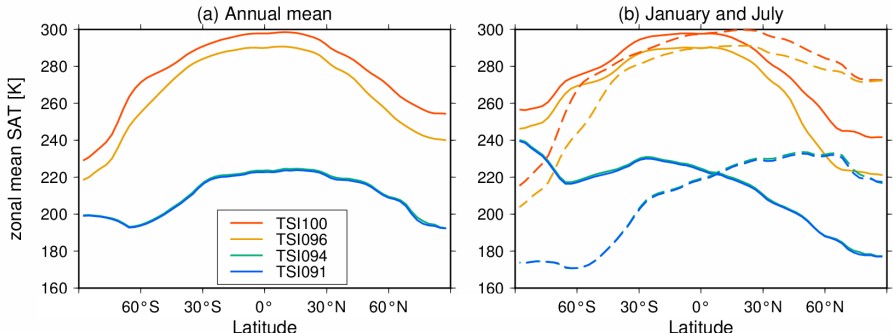

**Figure 10.** Zonal mean 2-m air temperature in the four experiments: (a) annual mean and (b) January mean (bold lines) and July mean (dashed lines)

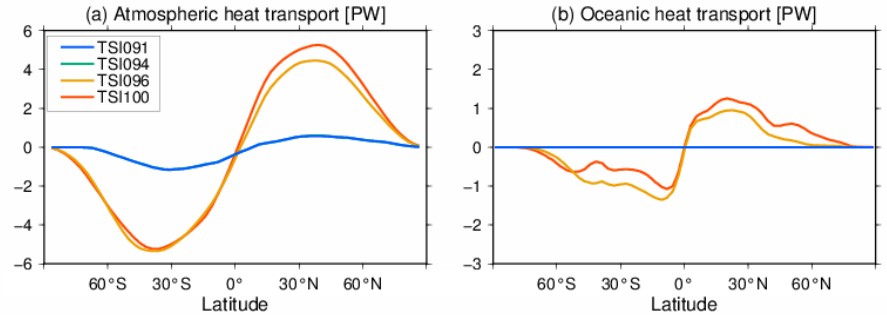

**Figure 11.** (a) Atmospheric and (b) oceanic meridional heat transport in the four experiments.

ceptibility of MIROC to the snowball climate in response to insolation change is the albedo of snow. In MIROC, the albedo of snow can be as high as 0.85 in a colder climate (Fig. 4b), which is near the maximum value adopted in the AOGCMs used in previous studies on modern snowball experiments (Yang et al., 2012a). We note that the snow albedo of 0.85 is the same as that used in the ICON-ESM (Ramme and Marotzke, 2022). Additionally, the global mean sea ice area before snowball onset was $\sim 40$ %, which is also smaller than that used in the other AOGCM studies listed above (i.e., 55–76 %). This indicates that MIROC tends to have less melting of sea ice in the mid-latitudes before snowball onset. One possible explanation is that the AABW cell used in the present-day simulation was stronger than that adopted in previous studies (Voigt and Marotzke, 2010), which might have reduced the temperature of the deep ocean and decreased the subsurface ocean temperature beneath the sea ice in the mid-latitudes.

## 4.2 Evolution of ocean circulation across snowball onset

At the end of the simulation up to 2000 years from the onset of the snowball state, active ocean circulations of approximately 20 Sv were maintained. The simulated meridional overturning cell was characterized as being symmetrical between the hemi-



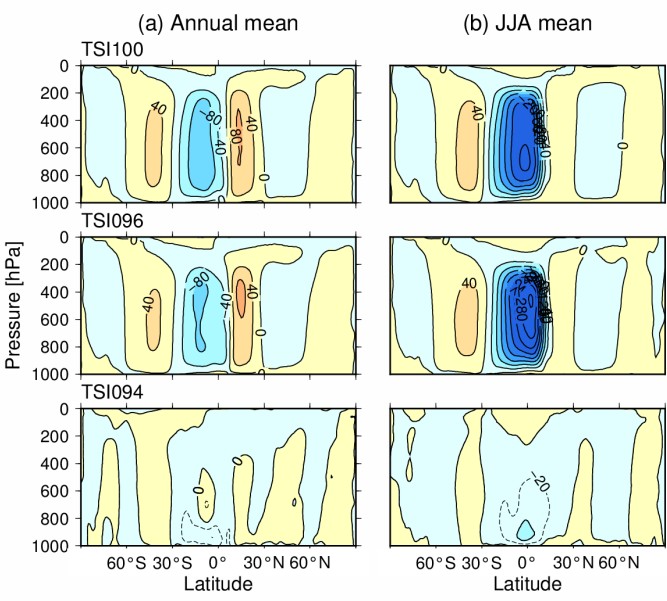

**Figure 12.** (a) Annual mean and (b) boreal summer meridional atmospheric mass streamfunction ($10^9 kg/s$, positive indicates clockwise circulation) in the three experiments.

spheres with upwelling near the equator (Fig. 5c), as found in one previous OGCM study that produced volume transport of

the same order of magnitude (Ashkenazy et al., 2014). Although our simulated MOC characterized by upwelling was similar to that in their study, the driving mechanisms of the meridional circulation would be different because geothermal heat flux was not included in our simulations. We speculate that the simulated meridional circulation in our model was driven by surface winds over the tropical ocean characterized by westward winds that induced upwelling over the equator (Fig. 6b). Our model setting enabled the effect of atmospheric winds to be transmitted to the ocean through the sea ice, even when the sea ice

became several hundred meters thick, as in the glaciation experiments of Ramme and Marotzke (2022). If we had conducted experiments in which the atmospheric momentum flux was not transmitted to the ocean under the thick sea ice, as in Pollard et al. (2017), the strong salinity stratification would have been maintained for a longer time.

Strong salinity stratification at snowball onset is present in the TSI094 experiment but absent in the TSI091 experiment, suggesting that the presence of salinity stratification before snowball onset depends on radiative forcing. If the external forcing

was strong and the climate turned into the snowball state in a relatively short time (TSI091), the strong stratification at snowball onset was absent. By contrast, if the transition to the snowball state was gradual, the strong salinity stratification could develop before snowball onset. This would explain why the salinity stratification was not observed in Ramme and Marotzke (2022), who found that it took approximately 150 years for snowball onset. It required several hundred years to resolve the salinity stratification in our TSI094 experiment (Fig. 8a left) than in TSI091 and the experiments of Ramme and Marotzke (2022).





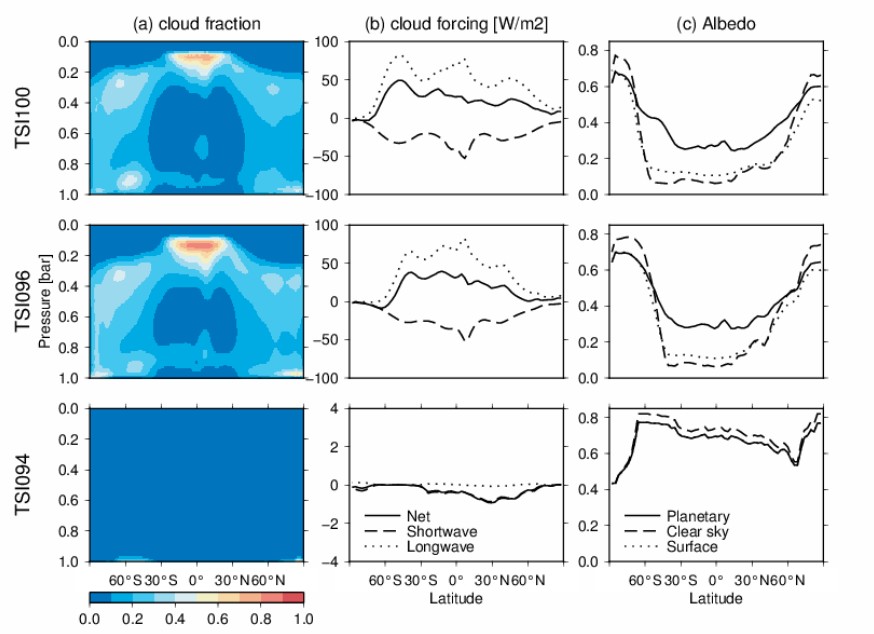

**Figure 13.** Zonal mean cloud fraction, cloud radiation effect for shortwave and longwave radiation, and zonal mean planetary albedo, clear sky albedo, and surface albedo in the TSI100, TSI096, and TSI094 experiments.

### 4.3 Potential limitation on ocean circulation

In the MIROC model, the surface momentum flux between the atmosphere and the ocean under the presence of sea ice is formulated using a nondimensional drag coefficient to relate local stress components at the ice–ocean interface, independent of sea ice thickness. Therefore, wind stress over the atmosphere–ice interface could induce movement of the sea ice and drive ocean currents just below the sea ice, thereby forcing the meridional overturning circulation (Fig. 5c). It is noted that this treatment might not be realistic in the snowball climate. For example, Pollard et al. (2017) set the ocean model to prevent sea ice advection by winds if the sea ice thickness exceeded 6 m, representing the transition from the modern sea ice state to that of the snowball Earth, as in the sea–glacier model coupled with the EBM (Goodman and Pierrehumbert, 2003; Pollard and Kasting, 2005; Tziperman et al., 2012). Our study, using the same configuration of sea ice dynamics as adopted for modern experiments, addressed the knowledge gap of how to formulate or parameterize the stresses and dynamics of thick sea ice over the ocean. Improvements in the model might include developments of a coupled ice sheet and ice shelf model as an extension of the Antarctic ice sheet and ice shelves. Other issues are related to the boundary conditions used in the climate model. For example, we did not apply geothermal heat flux in our experiments, but the sea ice thickness would reach a steady state of approximately 1000 m where the vertical diffusion of heat in the sea ice and the amount of geothermal heat flux are balanced. As shown by Ashkenazy et al. (2014), the upwelling ocean circulation at the equator is achieved via the inflow of geothermal energy and the thickness of the sea ice becoming thinner toward the equator. While the meridional overturning





circulation characterized by upwelling at the equator is common among experiments, the inclusion of geothermal heat flux and bathymetry can produce different results.

### 4.4 Potential limitation of atmosphere and additional sensitivity experiments

The simulated water cycle in the snowball climate in MIROC was substantially different from that of previous studies (Abbot et al., 2013), and this water cycle contradicted the geological record of glacial deposits over land. We found that net precipitation over the ocean (Fig. 14a left), which occurred through atmospheric vapor condensation at the sea ice surface, caused an extremely strong inversion layer to form only over the ocean (i.e., up to 20 °C difference between the 2-m air temperature and the surface temperature), which is not present in the modern simulation (Fig. 14b left). The prescribed ice sheet over land became the infinite source of atmospheric water vapor via sublimation of the ice sheet, which caused substantial evaporation from the land. MIROC tends to simulate more stable stratification in the atmospheric boundary layer in the present-day climate over Greenland, and this leads to excessive atmospheric vapor condensation during winter. This phenomenon occurred in our snowball simulation but with more extreme magnitude.

The treatment of the turbulence coefficient was not well constrained in the MIROC model and therefore we conducted additional experiments to change parameters related to the minimum turbulence coefficient in the atmospheric boundary layer over the ocean. In the sensitivity experiment TSI094mod, we increased the minimum value of the thermal diffusion coefficient over the ocean surface by 50 times to prevent an extremely strong atmospheric inversion. The TSI094mod experiment branched from the onset of the snowball climate (year 1330 in the TSI094 experiment), and we analyzed the final 50-year climatology of the 370-year simulation.

The TSI094mod experiment produced a weaker temperature inversion layer over the sea ice than that in TSI094 (Fig. 14b right), and the substantial atmospheric condensation that occurred only over the sea ice was markedly reduced. The value of net precipitation was approximately 0.5 mm per year (Fig. 14a right), which is in accord with the result derived from the FOAM model that is among the various multimodel AGCMs used for snowball simulations (Abbot et al., 2013). We note that the meridional atmospheric circulation in the snowball Earth climate (Fig. 12 bottom) was unchanged in TSI094mod. The results of additional experiments suggested that the improvement in the atmospheric boundary layer could have considerable impact on the climate of snowball Earth via the water cycle and ice sheet mass balance, and on the cloud fraction and its radiation effects regarding the required conditions for snowball termination. We also performed an experiment prescribing a surface albedo of 0.7, as adopted in previous multimodel studies (Abbot et al., 2013), by removing the temperature dependence of the albedo of snow and sea ice. We found that the derived results were closer to those of previous AGCM studies characterized by a stronger atmospheric circulation associated with seasonal cycles and net sublimation in the tropical region (not shown). This result indicates that the distribution of the surface albedo in our climate model (Fig. 4b) affects the atmospheric circulation and the cloud–radiation properties in the snowball state (Figs. 12 and 13 bottom). Although we found that the parameter of thermal diffusion coefficient over the ocean had little impact on the oceanic circulation during the snowball state, it could be crucial in quantifying the threshold of deglaciation from the snowball state (Pierrehumbert, 2004, 2005; Le Hir et al., 2007; Hu et al., 2011; Abbot et al., 2012, 2013; Abbot, 2014; Wu et al., 2021).





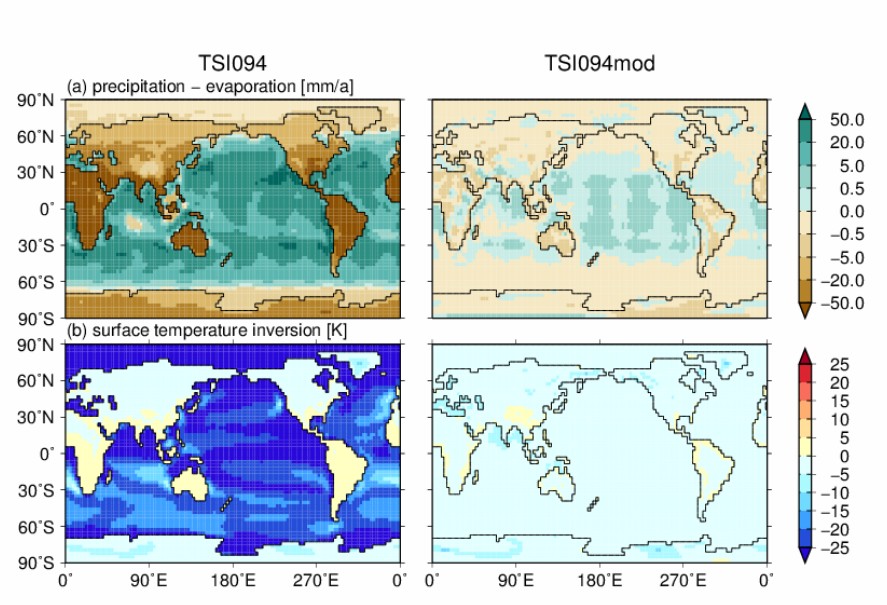

**Figure 14.** (a) Precipitation minus evaporation (includes ice sublimation) and (b) difference between surface temperature and 2-m air temperature in the TSI094 and TSI094mod experiments. In TSI094mod, the parameter of the minimum turbulence coefficient in the atmospheric boundary layer over the ocean was changed.

## 4.5 Other climate system feedbacks and implications

In the MIROC simulations, the extent of the continental ice sheet was assumed the same as that of the present-day in all simulations, meaning that the continental ice sheet was limited in Greenland and Antarctica. Although a standard AOGCM would not forecast changes in the ice sheet because such changes take much longer than the typical period of an AOGCM simulation, it has been shown that the prescription of the ice sheet can change the threshold for snowball onset (Liu et al., 2017). Donnadieu et al. (2003) and Pollard and Kasting (2004) used ice sheet models asynchronously coupled with an AGCM and found that the development of an ice sheet in low-latitude areas is possible because of net precipitation. Additionally, results derived from an ice sheet model coupled with the EBM indicated that the continental distribution affects the solution of the snowball state because of migration of ice sheet margins (Liu and Peltier, 2010). However, the simulated net precipitation was mostly negative over land, which means that it would be difficult to maintain a continental ice sheet over most continental areas if we coupled our climate model with an ice sheet model. We found that the water cycle is sensitive to the atmospheric boundary layer parameters (Fig. 14a); therefore, representation of the atmospheric water cycle must be improved to advance coupled climate–ice sheet simulations.

We note that several boundary conditions were different from the actual situation of snowball events during the Cryogenian Period. First, we used the modern configuration of the continental distribution. Previous studies showed that the reduction



in solar flux necessary for snowball onset was smaller than that associated with the present-day continental configuration, indicating that initiation of a snowball climate is easier if paleotopography is considered (Voigt et al., 2011; Liu et al., 2013). Second, we conducted simulations with incoming solar flux, but changes in atmospheric greenhouse gas forcing would produce different results because the distribution of radiative forcing would be different. Specifically, Liu et al. (2013) and Feulner and Kienert (2014) estimated the threshold of atmospheric $CO_2$ concentration at 80 and 150 ppm under Marinoan and Sturtian

configurations, respectively.

    Climate models prescribe atmospheric greenhouse gas concentrations as input parameters, but earlier three-dimensional OGCM studies showed that global cooling above the threshold of snowball onset contributes to oceanic carbon uptake and reduces the concentration of atmospheric $CO_2$ by 50–150 ppm (Oka et al., 2011; Liu et al., 2023), suggesting that the oceanic carbon cycle in response to cooling acts as positive feedback on snowball onset. The TSI094 results indicated that the ocean

can have very strong salinity stratification before snowball onset if the conditions of the external forcing are just below the threshold (Fig. 8a). In our simulation, the concentration of atmospheric $CO_2$ was set at a constant value during the 1000 years before snowball onset when the strong salinity stratification developed. If the model forecasts the marine carbon cycle and the concentration of atmospheric $CO_2$, the strong salinity stratification would uptake carbon in the deep and bottom ocean, as was found in both Oka et al. (2011) and Liu et al. (2023), thereby lowering the concentration of atmospheric $CO_2$ and possibly

causing snowball onset before the strong salinity stratification developed. It is also noted that the current continental distribution enables active sea ice production in the region of the Antarctic Circumpolar Current (Fig. 7), which might contribute to the formation of strong salinity stratification, and this should be examined using the past continental distribution. Additionally, the presence of ice sheets and the weathering rates associated with the past continental distribution could also cause changes in the concentration of atmospheric $CO_2$ (Tajika, 2003; Donnadieu et al., 2004; Peltier et al., 2007; Le Hir et al., 2009).

Despite the limitations in the models and experimental design adopted in this study, our simulations provide insights into the role of the oceanic circulation in the climate state before snowball onset, during snowball onset, and during a snowball Earth event. If the external forcing was slightly weaker than the threshold required for snowball onset, active sea ice production in the polar regions and associated brine rejection would increase the salinity of the deep ocean, while melting of sea ice in the mid–low latitudes would reduce the salinity of the surface ocean, resulting in strong stratification. This suggests that in intensive

glacial periods shorter than a snowball event, there would be strong ocean salinity stratification because of extensive sea ice cover. Once the forcing or concentration of atmospheric $CO_2$ associated with ocean carbon uptake exceeded the threshold for snowball onset, the entire globe would become covered in sea ice within a few decades, as reported in previous studies. After snowball onset, even with the current model settings of substantial air–sea flux, it would take several hundred years to eliminate the strong ocean stratification and to achieve a steady MOC with nearly uniform temperature and salinity fields. Our findings

highlight the necessity for evaluating the impact of ocean circulation changes on geochemical modeling, including continental ice sheet dynamics.



*Code and data availability.* The MIROC code associated with this study is available to those who conduct collaborative research with the model users under license from copyright holders. All model data supporting our findings will be archived at Zenodo.

*Author contributions.* TO and TK conceived the study. TO and TK improved the model for use in this study. TO, TK, and SS-T analyzed
the data. TO wrote the manuscript with input from all coauthors.

*Competing interests.* The authors declare that they have no competing interests.

*Acknowledgements.* We appreciate the fruitful discussions conducted with Joseph Kirschvink, Ryouta O'ishi, and Tatsuo Suzuki. This research is supported by JSPS Kakenhi 23K17709 and a "Strategic Research Projects" grant (No. 2024-SRP-03 (TK)) from the Research Organization of Information and Systems. TO receives additional support from JPMXD0722680395 and JSPS Kakenhi 24K17122. This
study is also supported by grants from the Astrobiology Center of the National Institutes of Natural Sciences (No. AB0608) and the Itoh Science Foundation (TK). The simulations were performed on the Earth Simulator 4 at the Japan Agency for Marine-Earth Science and Technology (JAMSTEC). We thank James Buxton MSc, from Liwen Bianji (Edanz) (www.liwenbianji.cn/), for editing the English text of a draft of this manuscript.




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
