# Peer review of "Climate and ocean circulation changes toward a modern snowball Earth"

_EGUsphere, 2025_

## Referee Comment (RC1)

The authors investigated the evolution of climate and ocean circulation towards a modern snowball Earth in MIROC4m, and found some phenomena that are quite different from those in previous studies, especially when the Earth has entered a hard snowball. I like the study in that their model can be continued for thousands of years after a snowball Earth has been initiated, which was not possible in CCSM3; the model would just crash in most cases (Yang et al., 2012; Liu et al., 2013). It is unclear whether this was achievable in ECHAM5/MPI-OM (Voigt and Marotzke, 2010). Stable snowball simulations are certainly possible in ICON-ESM (Ramme and Marotzke, 2022), but it has not been used to investigate in detail the evolution of oceanic and atmospheric circulation. Therefore, there is an opportunity to find something new from the simulations done by the authors using MIROC4m. However, to my understanding, the important phenomena found by the authors so far are mostly artefacts due to inappropriate settings in the model. These problems will be listed in detail below and because of which, I think they will have to redo some of the experiments and the corresponding analyses.

1. They found that there would be a strong nearly hemispherically symmetric MOC (Fig. 5c) even if the thickness of sea ice is more than 200 m thick (Fig. 4). I think this is due to the unrealistically large wind stress felt at the ice-ocean interface. This wind stress will cause large poleward Ekman transport off the equator and thus strong upwelling at the equator, which will drive strong and deep MOC when vertical stratification is absent. This MOC is an enlarged version of the wind-driven subtropical cell (called STC), which is only ~500 m deep under normal conditions (e.g., Fig. 3d). The authors recognized the limitation in this stress but probably did not realize how much their results would be affected by this drawback. I do not know under how thick sea ice the ocean should not feel the wind stress anymore, but sea ice of 200 m thick (like ice shelf around Antarctica) will certainly not move with wind anymore. Thus, they should better re-do the TSI094 and TSI091 simulations by fixing this limitation, in order to provide to the readers realistic results.

   When fixing the limitation on wind stress at the ice-ocean interface, it is probably also necessary to make the sea ice stagnant. This is still unrealistic (thick sea ice moves slowly) but would be better than the spatial distribution of ice thickness shown in their Fig. 4a (TSI094); one would expect that ice is thicker over the high latitudes than over the low latitudes as in Ashkenazy et al. (2014).

2. Another peculiar phenomenon they found is the net precipitation at the equator as well as the annual mean Hadley circulation that rises at the equator (I think the

two are related and can be considered as one). This phenomenon is opposite to what was found in the previous study by Abbot et al. (2013). However, this is also an artefact in my opinion because they set the land surface to be glacier once the ocean is completely covered by ice. When they do this, the land surface will have smaller surface albedo (Fig. 4b) and thus higher surface temperature (not shown but can be inferred). Moreover, the land surface will become an infinite source of water vapor. This is why the land surface has a strong net evaporation (Fig. 14) while the ocean has a net precipitation. A reasonable guess is that this also causes the air to rise over land and sink over the ocean, the latter will produce a strong temperature inversion over the ocean. Therefore, I do not feel that this temperature inversion should be attributed to the turbulent coefficient in the atmospheric boundary layer, and indeed, their test with a different coefficient could not remove the inversion (Fig. 14).

Another effect of the warm land surface is to shift the rising branch of the annual mean Hadley circulation to the north of equator, clearly seen in Fig. 12. Therefore, it is not actually a good idea to set the land surface as glacier when the sea ice closes off at the equator, just letting snow to accumulate on the land surface (i.e. do not do any special treatment) is probably more realistic. The authors do not need to worry about the glacier formation on the tropical lands, they will remain thin after even a few thousands of years because the net precipitation rate is small in a hard snowball Earth. That means, the authors need to redo the simulations without setting the land surface to glacier in order to show to the readers proper picture of atmospheric circulation.

3.  If the authors are willing to redo the simulations, the authors may want to look at how the snow cover changes with time over both land and sea ice; how the ocean stratification and MOC evolve, both the timescale and pattern could change significantly from those shown in the current manuscript; the gradual evolution of atmospheric circulation and how long it takes to reach equilibrium.

4.  Although land surface is assumed to become ice once the ocean is completely covered by sea ice, the land seems to have much lower albedo than the ocean. This is the major reason that the results here are very different from previous simulations. Abbot et al. (2013) did not include any continent explicitly, so they avoided this complexity. In your case, the snow is hard to accumulate on land, which creates a positive feedback that makes the land even warmer. If you prescribe the land as ice with thick snow, the results may look similar to previous

modeling results. I am not asking for more sensitivity test but something you can discuss.

5. Fig. 7 shows something interesting. Around year 1280, sea ice starts to grow near 30°S while the higher latitudes are still having a net melting. This process seems to trigger the runaway effect, why does this happen?

**Other Comments**

1) Please remove the statement about biogeochemical changes in the abstract as readers would expect much more from the manuscript by reading the abstract.
2) L16: "iron formation" to "banded iron formation" since iron formation could have many other forms and mechanisms.
3) L20: please explicitly state that the change of solar luminosity with time was estimated from solar models (i.e., not geological records) and provide relevant references.
4) L22: "Thus, it is …", I cannot see the logic from the context why "thus" should be used here.
5) L24: "modelingcan" -> "modeling can"
6) L38: "AOGCM" -> "AOGCMs"
7) L39-40: "reconstructions of the continental distribution" is repetitive
8) L57-59: In my opinion, although Ramme and Marotzke (2022) provided a nice demonstration of the ocean circulation during the snowball termination, the sea ice in their snowball state was quite thin so that the freshwater layer was easily eroded away. The simulations in Zhao et al. (2022) provide another perspective.
9) L98: is the shortwave albedo described here the mean of visible and near-infrared wave bands?
10) L99-101: Is snow aging considered over both land and sea ice in MIROC4m? This is important for explaining the different results between your model and other models since your model results seem peculiar. Also, I assume that the sea-ice albedo is thickness dependent and the value you provided represents the maximum.
11) L105-106: This is quite surprising because many models use sigma layers near the bottom because of its large variation in bathymetry and z coordinates near surface for the small variation of surface height.
12) Fig. 2: it will be useful to show the evolution of oceanic heat transport here; the shallow MOC is usually called STC (as mentioned above) in the field of physical oceanography; the deep MOC I guess, is driven by winds in the same way as STC,

which is fundamentally different from the deep MOC in TSI100 and TSI096. Although the deep MOC in the latter is also maintained by wind driven upwelling, it's sinking is due to density anomaly. The authors may want to make it clear whether the sinking is due to density anomaly or downward Ekman pumping.

13) Fig. 3: is there atmosphere-ocean heat exchange at year 1450-1499?

14) Fig. 5: what do the contours show in panel f.

15) L243&L332: I think Abbot et al. used an albedo of 0.6, why do the authors think it was 0.7?

16) L267-270: The explanation provided here is very unlikely. The ocean temperature below the permanent sea ice is always below 0°C whether the AABW cell is strong or not. This is clearly demonstrated in Fig. 2 of Yang et al. (2012). I would think the high snow albedo over sea ice is more likely the reason. This can be tested but I will not ask that much.

17) L289: A counterpart is missing for "than" in this sentence.

18) L291-293: there seems to be a grammatical error around "relate"

19) L302: "but" does not seem to be an appropriate conjunction word here.

20) L310: It should be more specific how your hydrological cycle contradicts with the geological record, not obvious to all readers.

21) L310-313: I don't understand why vapor condensation can induce a strong temperature inversion near the surface since condensation gives off large amount of heat to the surface.

22) Fig. 14: It will be useful to also show surface temperature here.

**Reference** (all the other references can be found from the reference list of the manuscript)

Zhao, Z., Y. Liu, H. Dai (2022), Sea-glacier retreating rate and climate evolution during the marine deglaciation of a snowball Earth, Global and Planetary Changes, 215, 103877, doi: 10.1016/j.gloplacha.2022.103877

---

## Author Comment (AC1)

We thank the two reviewers for providing precise and valuable feedback on our manuscript. We have conducted additional experiments to answer concerns raised by both reviewers. The reviewer's comments are indicated in black text, and our answers follow in blue text.

The first experiment aims to eliminate the influence of atmospheric winds on ocean circulation in the presence of very thick sea ice. We found that the recovery of the deep ocean circulation occurs in this experiment, and the final state is hemispherically symmetric meridional overturning circulations, similar to those in the standard experiment. This suggests that the thermal condition at the sea surface is the primary factor, and air-sea momentum flux plays a minor role in driving oceanic meridional overturning circulation.

The second set of experiments prescribes bare soil on the all land grid after the snowball onset instead of the ice sheet. We found it feasible to prescribe bare soil in all land grids by reducing the maximum albedo of snow. In contrast, the model crashes very early in the original maximum albedo of snow because of sustained atmospheric water vapour condensation over the sea ice. We have analysed the climate in the snowball simulations with bare land, and found that the similar atmospheric circulation and water cycle (stronger Hadley circulation, net sublimation in the equator and net precipitation in low latitudes) is consistent with previous studies.

We will be happy to integrate these experiments (eliminating air-sea flux in the hard snowball state, and prescribing bare soil over the land) into the main result and to revise the manuscript. Our complete response is below in this document.
* * *
Response to reviewer #1:

The authors investigated the evolution of climate and ocean circulation towards a modern snowball Earth in MIROC4m, and found some phenomena that are quite different from those in previous studies, especially when the Earth has entered a hard snowball. I like the study in that their model can be continued for

thousands of years after a snowball Earth has been initiated, which was not possible in CCSM3; the model would just crash in most cases (Yang et al., 2012; Liu et al., 2013). It is unclear whether this was achievable in ECHAM5/MPI-OM (Voigt and Marotzke, 2010). Stable snowball simulations are certainly possible in ICON-ESM (Ramme and Marotzke, 2022), but it has not been used to investigate in detail the evolution of oceanic and atmospheric circulation. Therefore, there is an opportunity to find something new from the simulations done by the authors using MIROC4m. However, to my understanding, the important phenomena found by the authors so far are mostly artefacts due to inappropriate settings in the model. These problems will be listed in detail below and because of which, I think they will have to redo some of the experiments and the corresponding analyses.

Thank you for carefully reading and providing us with valuable comments. We have addressed concerns about model settings and experimental designs by performing additional experiments. We are pleased to utilise these new experiments as the primary results in the revised manuscript. The details of the experiments and results are in our point-by-point replies.

1. They found that there would be a strong nearly hemispherically symmetric MOC (Fig. 5c) even if the thickness of sea ice is more than 200 m thick (Fig. 4). I think this is due to the unrealistically large wind stress felt at the ice-ocean interface. This wind stress will cause large poleward Ekman transport off the equator and thus strong upwelling at the equator, which will drive strong and deep MOC when vertical stratification is absent. This MOC is an enlarged version of the wind-driven subtropical cell (called STC), which is only ~500 m deep under normal conditions (e.g., Fig. 3d). The authors recognized the limitation in this stress but probably did not realize how much their results would be affected by this drawback. I do not know under how thick sea ice the ocean should not feel the wind stress anymore, but sea ice of 200 m thick (like ice shelf around Antarctica) will certainly not move with wind anymore. Thus, they should better re-do the TSI094 and TSI091 simulations by fixing this limitation, in order to

provide to the readers realistic results. When fixing the limitation on wind stress at the ice-ocean interface, it is probably also necessary to make the sea ice stagnant. This is still unrealistic (thick sea ice moves slowly) but would be better than the spatial distribution of ice thickness shown in their Fig. 4a (TSI094); one would expect that ice is thicker over the high latitudes than over the low latitudes as in Ashkenazy et al. (2014).

We agree that the original experimental design, which involved wind stress felt at the ice-ocean interface in the hard snowball is an extreme setting. We conducted additional experiments branched at year 1300 of the TSI094, with excluding wind stress on the ocean (Experiment Name TSI094NWS, NWS indicates No Wind Stress to the ocean). In this setting, the sea ice is not completely stagnant; it can still move due to ocean circulation. We conducted another 670-year simulation to reach 2,000 years from the start of the simulations.

In the TSI094NWS case, the deep MOC resumes after 500 years of the snowball onset (Figure R1-1). At the final state, the hemispherically symmetric MOC is developed (Figure R1-2 right panel). This result indicates that the air-sea momentum flux is not the necessary driver of the hemispherically symmetric overturning circulation. Our discussion in the original manuscript, "*We speculate that the simulated meridional circulation in our model was driven by surface winds over the tropical ocean characterised by westward winds that induced upwelling over the equator (L277-278)*", was not true. Therefore, this statement will be amended in the revised manuscript. The TSI094 experiment accelerates the recovery of the deep MOC compared to the TSI094NWS experiment, suggesting that the sea surface momentum flux in TSI094 aids in the recovery of the deep MOC following snowball onset.

Based on the new results, we now think the driver of the hemispherically symmetric overturning circulation is the difference in the sea surface salinity flux. The greater sea ice production in high latitudes than in the low latitudes in the snowball climate (Fig. 7), which induced the density flux at the sea surface and ultimately drives the MOC (Fig. R1-2 middle and right panels). The thickness of sea ice exhibits a less pronounced zonal gradient than the original TSI094 (Fig.

R1-3), which is more similar to the sea-glacier model (Ashkenazy et al., 2014).

[Figure]

Figure R1-1: The timeseries of the deep MOC transport (exact definition as Figure 2 of the manuscript. The black line represents the new experiment TSI094NWS, where sea surface momentum flux is set to zero after snowball onset.

[Figure]

Figure R1-2: (top panels) meridional overturning circulation [Sv]. (bottom) zonal mean salinity. The figure is in the same format as Figure 5 of the manuscript.

[Figure]

Figure R1-3: sea ice thickness of the TSI094 and TSI094NWS experiments

2. Another peculiar phenomenon they found is the net precipitation at the equator two are related and can be considered as one). This phenomenon is opposite to what was found in the previous study by Abbot et al. (2013). However, this is also an artefact in my opinion because they set the land surface to be glacier once the ocean is completely covered by ice. When they do this, the land surface will have smaller surface albedo (Fig. 4b) and thus higher surface temperature (not shown but can be inferred). Moreover, the land surface will become an infinite source of water vapor. This is why the land surface has a strong net evaporation (Fig. 14) while the ocean has a net precipitation. A reasonable guess is that this also causes the air to rise over land and sink over the ocean, the latter will produce a strong temperature inversion over the ocean. Therefore, I do not feel that this temperature inversion should be attributed to the turbulent coefficient in the atmospheric boundary layer, and indeed, their test with a different coefficient could not remove the inversion (Fig. 14).

Another effect of the warm land surface is to shift the rising branch of the annual mean Hadley circulation to the north of equator, clearly seen in Fig. 12. Therefore, it is not actually a good idea to set the land surface as glacier when the sea ice closes off at the equator, just letting snow to accumulate on the land surface (i.e. do not do any special treatment) is probably more realistic. The authors do not need to worry about the glacier formation on the tropical lands, they will remain thin after even a few thousands of years because the net precipitation rate is small in a hard snowball Earth. That means, the authors need to redo the simulations without setting the land surface to glacier in order to show to the readers proper picture of atmospheric circulation.

We agree with the points that net precipitation and Hadley circulation are very different from the previous study by Abbot et al. (2013), as discussed in the manuscript (section 4.4). We have conducted another set of additional experiments to investigate the cause of the net precipitation pattern further. All additional experiments are branched at the year 1330 of TSI094, corresponding

to the snowball onset.

Experiment 1: The entire land surface is replaced with bare soil instead of an ice sheet. We found there was still substantial atmospheric condensation as in the original TSI094 experiment, and the model crashed within one year because atmospheric water vapor condensation led to global atmospheric water vapor becoming negative. This result indicates that the substantial sublimation from the land ice sheet is not the cause but the result of water vapor condensation over sea ice. Our original experimental design prescribing land as an ice sheet prevents model crashes in a hard snowball condition.

Experiment 2 (TSI094bareland): The all land surface is replaced with the bare soil instead of ice sheet. Furthermore, the maximum albedo of the snow is reduced (0.85 to 0.7 in this response letter experiment) to prevent substantial water vapor condensation over sea ice. We found that there is net sublimation near the equator and positive net precipitation in the low-latitudes (Fig. R1-4), consistent with previous studies (Abbot et al. 2013). And TSI094bareland exhibits a strong seasonal Hadley circulation (Fig. R1-5 right panels), consistent with previous studies.

We find that prescribing all land as bare land and reducing snow albedo has another effect that the surface air temperature is much warmer than the original experiment (-34℃ in the global mean surface air temperature, compared to -58℃ in TSI094). In substantial land areas, the summer snow cover can be zero and bare land is exposed. The appearance of bare land in snowball climate can be consistent with previous studies with GCM and ice sheet models (Pollard and Kastings 2004; Benn et al. 2015. The experiments shown in the response letter use an ad hoc setting to prescribe the maximum snow albedo as 0.7, but in the revised manuscript, we plan to cover the albedo setting and the impact of the uncertainty in the deep ocean circulation.

[Figure]

Figure R1-4: (Left) zonal mean net precipitation net precipitation in the TSI094 and TSI094bareland experiments, respectively. (Right) map of net precipitation in the TSI094bareland experiment.

[Figure]

Figure R1-5: Hadley circulations in the TSI094 and TSI094bareland experiments, respectively.

3. If the authors are willing to redo the simulations, the authors may want to look at how the snow cover changes with time over both land and sea ice; how the ocean stratification and MOC evolve, both the timescale and pattern could change significantly from those shown in the current manuscript; the gradual evolution of atmospheric circulation and how long it takes to reach equilibrium.

Thank you for your suggestion. We have analyzed land snow cover changes in the TSI094bareland experiments at the three time slices (Figure 1-6. The three time slices are the same as Figure 3). The land snow cover advances to low latitudes, but there was substantial area without summer snow cover even before

and after the snowball onset, since summer melting of snow exceeds snowfall.

[Figure]

Figure R1-6: Sea ice extent and summer land snow cover (red area indicates snow cover) at three selected time-slice snapshots from the TSI094 experiment

4. Although land surface is assumed to become ice once the ocean is completely covered by sea ice, the land seems to have much lower albedo than the ocean. This is the major reason that the results here are very different from previous simulations. Abbot et al. (2013) did not include any continent explicitly, so they avoided this complexity. In your case, the snow is hard to accumulate on land, which creates a positive feedback that makes the land even warmer. If you prescribe the land as ice with thick snow, the results may look similar to previous modeling results. I am not asking for more sensitivity test but something you can discuss.

Thank you for the suggestion. We agree that the lower albedo of the ocean contributes to the warmer surface temperature and the sublimation of ice, which makes further positive feedback. In the additional experiment used in the revised manuscript (TSI094bareland), the reduced maximum albedo of snow (over the sea ice) reduces the contrast in the surface albedo between the ocean and land, which finally reduces the positive feedback above. We will also discuss how the presence of a continent increases complexity compared to simulations with aqua-planet or land-planet configurations.

5. Fig. 7 shows something interesting. Around the year 1280, sea ice starts to grow near 30°S while the higher latitudes are still having a net melting. This process seems to trigger the runaway effect, why does this happen?

Thank you for the suggestion. Precisely, the net sea ice production is occurring

near 30S around year y1280. According to the map of net sea ice production at this time slice, the sea ice was mostly positive at around this latitudinal band and there were no hot spots in sea ice production. One explanation is that 30S is associated with oceanic circulation characterized by divergence of sea ice.

Other Comments

1) Please remove the statement about biogeochemical changes in the abstract as readers would expect much more from the manuscript by reading the abstract.

Thank you for the suggestion. We will end the abstract with statements on the physics of the atmosphere and ocean.

2) L16: "iron formation" to "banded iron formation" since iron formation could have many other forms and mechanisms.

We change to "banded iron formation".

3) L20: please explicitly state that the change of solar luminosity with time was estimated from solar models (i.e., not geological records) and provide relevant references.

We clarify that solar luminosity was estimated from solar models with references.

4) L22: "Thus, it is …", I cannot see the logic from the context why "thus" should be used here.

We change the sentences:

"were changing over time and differed from the current situation. It is one outstanding question why snowball Earth events occurred during the Paleoproterozoic Era and the Cryogenian Period, but not since"

5) L24: "modelingcan" -> "modeling can"
6) L38: "AOGCM" -> "AOGCMs"

We correct them

7) L39-40: "reconstructions of the continental distribution" is repetitive

We correct as follows:

Subsequent studies used the same AOGCM to investigate the threshold for snowball onset under the past configuration of …

8) L57-59: In my opinion, although Ramme and Marotzke (2022) provided a nice demonstration of the ocean circulation during the snowball termination, the sea ice in their snowball state was quite thin so that the freshwater layer was easily eroded away. The simulations in Zhao et al. (2022) provide another perspective.

Thank you for telling us the reference. We will refer to Zhao et al. (2022) in the introduction, which estimates the quantitative timescale of snowball termination and the evolutions in ocean circulation.

9) L98: is the shortwave albedo described here the mean of visible and near-infrared wave bands?

We correct the manuscript by clarifying the visible and near-infrared albedo values.

"The shortwave (visible/near-infrared) albedo of snow was defined as a function of temperature to parameterize partial snow cover at relatively high temperatures, i.e., the albedo was set at a value of (0.85/0.65) for temperatures of −5 ◦C or colder, and it was reduced linearly to a value of (0.65/0.5) for temperatures up to 0 ◦C"

10) L99-101: Is snow aging considered over both land and sea ice in MIROC4m? This is important for explaining the different results between your model and other models since your model results seem peculiar. Also, I assume that the sea-ice albedo is thickness dependent and the value you provided represents the maximum.

No, snow aging is not considered in our model in both land and sea ice. The albedo of snow depends only on temperature.

Also, the albedo of the sea ice is independent of thickness. Sea surface albedo

is determined by sea ice concentration and the presence of snow over the sea ice. We clarify in the revised manuscript.

11) L105-106: This is quite surprising because many models use sigma layers near the bottom because of its large variation in bathymetry and z coordinates near surface for the small variation of surface height.

The choice of vertical coordinate is the same as the original ocean model, COCO, which focuses on general ocean circulation. One merit of using sigma near the surface is that it explicitly calculates the spatial distribution of dynamic sea surface height. The flow of the ocean near the bottom, one limitation of the z coordinate model, is parameterised with a bottom boundary layer scheme.

12) Fig. 2: it will be useful to show the evolution of oceanic heat transport here; the shallow MOC is usually called STC (as mentioned above) in the field of physical oceanography; the deep MOC I guess, is driven by winds in the same way as STC, which is fundamentally different from the deep MOC in TSI100 and TSI096. Although the deep MOC in the latter is also maintained by wind driven upwelling, it's sinking is due to density anomaly. The authors may want to make it clear whether the sinking is due to density anomaly or downward Ekman pumping.

Thank you for the suggestion. While the time series of oceanic meridional circulation in the TSI094 is shown in Figure 9 with latitudinal distribution, we are going to add the time series of maximum meridional heat transport in all simulations in Figure 2.

Also, we will discuss that the result of the additional experiment (TSI094NWS, no wind stress) clarify that the sinking due to density anomaly is the primary factor driving the deep MOC.

13) Fig. 3: is there atmosphere-ocean heat exchange at year 1450-1499?

Yes, there is heat exchange between the atmosphere and the ocean in the year 1450-y1499, and this thermal heat flux is used in sea ice growth around 0.1-0.3

m/a (Figure 7).

14) Fig. 5: what do the contours show in panel f.

The contour in panel f indicates a salinity contour interval of 0.05 psu, a much smaller interval than panels d-e (0.2 psu). We will remove them in the revision to avoid confusion and to focus on the fact that the ocean salinity is mostly uniform in TSI094.

15) L243&L332: I think Abbot et al. used an albedo of 0.6, why do the authors think it was 0.7?

Our experiment of prescribing a uniform albedo of 0.7 was conducted independently of the Abbot et al. papers. We correct the sentence to show that the albedo value is not identical to the Abbot et al. papers, which used a uniform albedo of 0.6. We are going to use the new experiment prescribing bare land over the land (TSI094bareland) in the main results to address Hadley circulations.

16) L267-270: The explanation provided here is very unlikely. The ocean temperature below the permanent sea ice is always below 0°C whether the AABW cell is strong or not. This is clearly demonstrated in Fig. 2 of Yang et al. (2012). I would think the high snow albedo over sea ice is more likely the reason. This can be tested but I will not ask that much.

We agree that the high snow albedo over sea ice is a likely reason. However, the efficiency in the meridional heat transport associated with the strength of AABW can enhance the response to the insolation forcing. We will revise the sentence to focus on the meridional heat transport by the ocean, not the strength of AABW.

17) L289: A counterpart is missing for "than" in this sentence.

We correct the sentence:

It required more several hundred years to resolve the salinity stratification in our TSI094 experiment (Fig. 8a left) than in TSI091 and the experiments of Ramme and Marotzke (2022).

18) L291-293: there seems to be a grammatical error around "relate"

We corrected the grammatical error:

… the surface momentum flux between the atmosphere and the ocean under the presence of sea ice is formulated using a nondimensional drag coefficient which is independent of sea ice thickness.

19) L302: "but" does not seem to be an appropriate conjunction word here.

We corrected it by dividing it to two sentences:

For example, we did not apply geothermal heat flux in our experiments. Based on the balance between the vertical diffusion of heat in the sea ice and the typical value in the geothermal heat flux of the Earth, the sea ice thickness would reach approximately 1000 m in a steady state.

20) L310: It should be more specific how your hydrological cycle contradicts with the geological record, not obvious to all readers.

We improve the sentences:

…this water cycle contradicted the geological record of glacial deposits in the coastal area in the low latitudes, suggesting the presence of an ice sheet in a substantial area of the Earth (Kirschvink et al., 1992; Hoffman et al., 2017).

21) L310-313: I don't understand why vapor condensation can induce a strong temperature inversion near the surface since condensation gives off large amount of heat to the surface.

We revised the sentence to clarify that the atmospheric vapor condensation is a result of a strong temperature inversion. A similar strong atmospheric temperature inversion occurs in the pre-industrial simulation over Greenland, but with slightly different rates.

We found that net precipitation over the ocean (Fig. 14a left) is associated with atmospheric vapor condensation at the sea ice surface, which is caused by an extremely strong inversion layer at the sea surface (i.e., up to 20 ◦C difference

between the 2-m air temperature and the surface temperature). A similar strong atmospheric temperature inversion occurs in the pre-industrial simulation over Greenland, but with slight rates.

22) Fig. 14: It will be useful to also show surface temperature here.
We agree with adding surface temperature in Figure 14. In the original (Fig. R1-6 left), the minimum turbulence coefficient in the atmospheric boundary layer over the ocean increases surface temperature because it weakens the temperature inversion layer. The figure R1-6 also shows that the presence of land albedo feedback in the original TSI094 experiment (discussed in major comment #4) and the TSI094mod substantially reduces the contrast in surface temperature between land and ocean.
We will change the experiment name TSI094mod to a new name to clarify that the minimum turbulence coefficient has been modified.

[Figure]

Fig R1-7: annual-mean skin temperature in the TSI094 and TSI094mod experiments.

Response to reviewer #2:
The manuscript by Takashi Obase and colleagues presents simulation results from climate experiments of transitioning from a modern-day climate state to a snowball-Earth state in response to an abrupt reduction in incoming solar radiation. The authors use the MIROC4m atmosphere–ocean general circulation model (AOGCM), which—in principle—seems a good choice for this kind of study and enables them to investigate the evolution of ocean and atmosphere during

and after that transition. I think the study is interesting especially with regard to the ocean dynamics and the results are discussed in an enlightening way. However, I would suggest a few general and a number of specific minor revisions before publication.

Thank you for carefully reading and providing us with valuable comments. We have addressed the concerns of model settings and experimental designs by performing additional experiments. We are pleased to utilise these new experiments as the primary results in the revised manuscript. The details of the experiments and results are in our point-by-point replies.

Minor revisions (general)

The authors acknowledge that their results differ from previous studies in some important points, which is not a problem in its own right, of course. However, two assumptions appear very critical to me, as outlined below. Ideally, a study would mostly cover the sensitivity of the results to these assumptions, but seeing that the scope of the paper would considerably grow and the discussed results do not need to be "most realistic", I would suggest an even deeper discussion of the following points.

   I think it is a strong assumption and maybe a large perturbation for the model to abruptly replace all land surface with ice sheets once the oceans are fully covered in sea ice. The effect of this assumption on the results (especially regarding the hydrological cycle) should be discussed.

We agree with the points that net precipitation and Hadley circulation are very different from previous study by Abbot et al. (2013), as already discussed in the manuscript (section 4.4). We have conducted another set of additional experiments to investigate the cause of the net precipitation pattern further. All additional experiments are branched at the year 1330 of TSI094, corresponding to the snowball onset.

Experiment 1: The entire land surface is replaced with bare soil instead of an ice sheet. We found there was still substantial atmospheric condensation as in the original TSI094 experiment, and the model crashed within one year

because atmospheric water vapor condensation led to global atmospheric water vapor becoming negative. This result indicates that the substantial sublimation from the land ice sheet is not the cause but the result of water vapor condensation over sea ice. Our original experimental design prescribing land as an ice sheet prevents model crashes in a hard snowball condition.

Experiment 2 (TSI094bareland): The all land surface is replaced with the bare land instead of ice sheet. Furthermore, the maximum albedo of the snow is reduced (0.85 to 0.7 in this response letter experiment) to prevent substantial water vapor condensation over sea ice. We found that there is net sublimation near the equator and positive net precipitation in the low-latitudes (Fig. R1-4), consistent with previous studies (Abbot et al. 2013). And TSI094bareland exhibits a strong seasonal Hadley circulation (Fig. R1-5 right panels), consistent with previous studies.

We find that prescribing all land as bare land and reducing snow albedo has another effect that the surface air temperature is much warmer than the original experiment (-34℃ in the global mean surface air temperature, compared to -58℃ in TSI094). In substantial land areas, the summer snow cover can be zero and bare land is exposed. The appearance of bare land in snowball climate can be consistent with previous studies with GCM and ice sheet models (Pollard and Kastings 2004; Benn et al. 2015. The experiments shown in the response letter use an ad hoc setting to prescribe the maximum snow albedo as 0.7, but in the revised manuscript, we plan to cover the albedo setting and the impact of the uncertainty in the deep ocean circulation.

[Figure]

Figure R2-1: (Left) zonal mean net precipitation net precipitation in the TSI094 and TSI094bareland experiments, respectively. (Right) map of net precipitation in the TSI094bareland experiment.

[Figure]

Figure R2-2: Hadley circulations in the TSI094 and TSI094bareland experiments, respectively.

The comparatively strong momentum transfer from the wind to the ocean in the case of very thick sea ice is already discussed. Ideally, one would need a sensitivity run to see the differences in the model. This is not absolutely necessary from my point of view, but it should be stressed at a prominent point of the manuscript that the assumption is probably unrealistic.

We agree that the original experimental design, which involved wind stress felt at the ice-ocean interface in the hard snowball is an extreme setting. We conducted additional experiments branched at year 1300 of the TSI094, with excluding wind stress on the ocean (Experiment Name TSI094NWS, NWS indicates No Wind Stress to the ocean). In this setting, the sea ice is not completely stagnant; it can still move due to ocean circulation. We conducted another 670-year simulation to reach 2,000 years from the start of the simulations.

In the TSI094NWS case, the deep MOC resumes after 500 years of the snowball onset (Figure R1-1). At the final state, the hemispherically symmetric MOC is developed (Figure R1-2 right panel). This result indicates that the air-sea momentum flux is not the necessary driver of the hemispherically symmetric

overturning circulation. Our discussion in the original manuscript, "*We speculate that the simulated meridional circulation in our model was driven by surface winds over the tropical ocean characterised by westward winds that induced upwelling over the equator (L277-278)*", was not true. Therefore, this statement will be amended in the revised manuscript. The TSI094 experiment accelerates the recovery of the deep MOC compared to the TSI094NWS experiment, suggesting that the sea surface momentum flux in TSI094 aids in the recovery of the deep MOC following snowball onset.

Based on the new results, we now think the driver of the hemispherically symmetric overturning circulation is the difference in the sea surface salinity flux. The greater sea ice production in high latitudes than in the low latitudes in the snowball climate (Fig. 7), which induced the density flux at the sea surface and ultimately drives the MOC (Fig. R1-2 middle and right panels). The thickness of sea ice exhibits a less pronounced zonal gradient than the original TSI094 (Fig. R1-3), which is more similar to the sea-glacier model (Ashkenazy et al., 2014).

[Figure]

Figure R2-3: timeseries of the deep MOC transport (same definition of Figure 2 of the manuscript. The black line represents the new experiment TSI094NWS, where sea surface momentum flux is set to zero after snowball onset.

[Figure]

Figure R2-4: (top panels) meridional overturning circulation [Sv]. (bottom) zonal mean salinity. The figure is same format as Figure 5 of the manuscript.

[Figure]

Figure R2-5: sea ice thickness of the TSI094 and TSI094NWS experiments

That said, it is notable that the authors included a discussion and sensitivity test regarding the minimal thermal diffusion coefficient over the ocean surface, because this is another critical point.

Thank you. The atmospheric circulations in the snowball climate is now addressed with one experiment with prescribing bare land (TSI094bareland).

We will retain the TSI094mod (which minimally changes the thermal diffusion coefficient over the ocean) in the main result, as it has a minimal impact on the pre-industrial simulation (TSI100).

Minor revisions (specific)

L8: I find it not clear what is meant by "necessary". Maybe replace the formulation by "the duration between the change in solar constant and snowball onset" or something similar.

We revise the sentence:

"By contrast, such salinity stratification was absent if the duration between the change in solar constant and snowball onset was short."

L17f.: "iron formation" is too general.

We changed it as "banded iron formation".

L20: "… 94% of its …" without "that"

L27: "… equilibrium solution of the climate" instead of "planet"

We correct them.

L38: "… the same AOGCM[s]" is not correct. E.g., the study by Poulsen et al. (2002) used FOAM, which the studies mentioned before did not employ. Voigt (2013) used an AGCM. The study by Eberhard et al. (2023) differs from the ones mentioned before, as they used a model of intermediate complexity (which is not an AOGCM) and focused on a large set of sensitivity runs instead.

We agree with this. As the focus of this sentence is on previous studies that have examined the role of continental distribution, we remove the phrase "the same AOGCMs" from this sentence. The phrase "same AOGCM" would appear in the subsequent sentences to compare the threshold of solar flux for the onset of a snowball onset between different continental distributions.

L39: The terms "Marinoan" and "Sturtian" have to be switched. The Marinoan is the younger one.

We correct the mistake: "Sturtian (720–660 Ma) and Marinoan (650–630 Ma)"

L65 and other instances: It appears that sometimes the names MIROC and MIROC4m are synonymous. Are they? If not, please make a clearer distinction.

MIROC and MIROC4m are used interchangeably in this article. We unify the wording to "MIROC4m" throughout the manuscript.

L95: I think this value of the ECS is for doubling CO2 in a modern climate state, right? If so, please mention this, as for states with more ice the ECS is sometimes expected to be much higher.
Yes, the ECS is defined in the modern, pre-industrial climate. The ECS can change depending on the reference climate. We mention it in the revised manuscript.

L112f.: Is it likely that this nonconservation of global water volume and salinity introduces any artificial long-term salinity trends for "stable" climate states which could be relevant for the snowball simulations? One could, e.g., check the discussed TS100 run for the long-term evolution of total salinity. I ask because sometimes this can happen in the case of nonconservative schemes.
We have checked the global-mean salinity trends in TSI100 experiment, and found the global-mean salinity trend is 0.002 psu per 1000 years, which is of the same magnitude as the pre-industrial simulations and substantially smaller than the global-mean salinity (34.71 in TSI100 experiment)

L120: What is the reason for using this value for the solar constant? Basically, I am wondering where the .12 comes from.
The solar constant value (1366.12) is the same as the description paper of the MIROC4m (Nozawa et al., 2007), which was submitted to the Fourth phase of the Coupled Model Intercomparison Project (CMIP4). We opted to use the same value because all model tuning for the pre-industrial simulation has been conducted with this solar constant.

L120: "changed from 91 to 100%" or "set to 91–100%"
We correct it.

L135ff.: I realize the difficulty to run the TS096 simulation for even longer, but it is not evident for me that this one will stay in a non-snowball state forever. Sea-ice area and SAT still have considerable trends and might reach the transition point eventually. I suggest to at least mention this in the manuscript and, for example, weaken the statement in L139f: "… was determined to be between 94 and 96%".

We agree that the equilibrium state of TSI096 may be a snowball state, given the gradual trend observed at the end of the simulation. We change the sentences to clarify that the 94% is a sufficient condition: "There are still gradual trends in sea ice area in the TSI096 simulations, and we don't rule out that the true equilibrium state of TSI096 is snowball. Therefore, the solar flux of 94 % is a sufficient condition for a modern snowball climate in MIROC4m."

L155: "Fig. 2c blue"—Please check whether you indicated the right color.
We correct it as "Figure 2c green" because the sentence "strength of the AABW cell remained at approximately 50 Sv" indicates TSI094 result.

L190f.: "The sea ice formation …" is a repetition of a similar statement in L184ff.
Agree with this. We changed it as "The increase in total sea ice volume contributes to the increase in global mean salinity".

L192: It would make sense to add a reference to Fig. 5c, as well.
We revise the sentence by referring to Figure 5c as well as Figure 5f.

L196: "maximum sea ice thickness along the western side of the Pacific Ocean"— This is difficult to see in the figure.
The sea ice thickness distribution (TSI094) with colour contours is displayed in Figure R2-5, which shows that sea ice thickness along the western side of the Pacific Ocean is greater than that along the eastern side, due to meridional transport by winds. The revised sentence will focus on that sea ice thickness is maximum in high latitudes and minimum in the low latitudes.

L203f.: This last sentence is partly a repetition of a similar earlier statement in L199f.

We agree with this. However, as we will be using TSI094NWS (no wind stress) in the main result, the statement about wind will be removed, instead focusing on salinity flux driving ocean circulations.

L246: ".. from that in a multimodel study", similar in L249f.

We correct it:

L284: "… depends on radiative forcing"—This sounds as if the radiative forcing directly influenced the salinity stratification. Maybe: "depends on the rate of cooling"?

We agree with this, corrected: "suggesting that the presence of salinity stratification before snowball onset depends on the rate of cooling"

L284f.: Please specify what you mean with "external forcing", this could even be early in the paper. Some people describe the insolation itself as a forcing, and then it would sound strange to speak of a stronger external forcing for reduced insolation.

As the focus of this sentence is the duration required for snowball onset, we revise the sentence without using external forcing: "If the amplitude of external forcing was strong"
Also, we add the wording "external forcing" in the method section. We clarify that the present study assumes solar flux represents the external forcing.

L289: "as opposed to" instead of "than"?

We change it as suggested: in our TSI094 experiment (Fig. 8a left) as opposed to TSI091 and the experiments of Ramme and Marotzke (2022)

L347: "… coupled with an EBM"

We correct it

L359: Please carefully revise the values given. Liu et al. (2013) find thresholds between 80 and 150 ppm, but thresholds even below 80 ppm for another aerosol parametrization. Feulner and Kienert (2014) find thresholds of 100–110 ppm for the Sturtian and 120–130 ppm for the Marinoan.

We carefully revise the sentences according to each reference:

Specifically, Liu et al. (2013) estimated the threshold of atmospheric CO2 as 80 ppm for the Sturtian and 150 ppm for the Marinoan configuration, and it can change by ~30 ppm depending on the optical depth of the aerosol parameterisation. Feulner Kienert (2014) estimated the threshold of atmospheric CO2 concentration as 110 for Sturtian and Marinoan configurations.

References

Nozawa, T., Nagashima, T., Ogura, T., Yokohata, T., Okada, N., Shiogama, H., Climate Change Simulations with a Coupled Ocean-Atmosphere GCM Called the Model for Interdisciplinary Research on Climate MIROC, CGER's Supercomputer Monograph Report, Vol. 12, (2007).
URL: https://www.cger.nies.go.jp/publications/report/i073/I073.pdf